# Developing the digital transformation skills framework: A systematic literature review approach

**Machiel Bouwmans**[1,2]*, **Xander Lub**[1,3], **Marissa Orlowski**[4,5], **Thuy-Vy Nguyen**[2]

**1** Research Group Organizations in Digital Transition, HU Utrecht University of Applied Sciences, Utrecht, The Netherlands, **2** Institute for People & Business, HU Utrecht University of Applied Sciences, Utrecht, The Netherlands, **3** Research Group Strategy, Organization & Leadership, Nyenrode Business University, Breukelen, The Netherlands, **4** MV Hospitality Solutions, LLC, Aurora, Colorado, United States of America, **5** Department of Marketing, Entrepreneurship, Hospitality and Tourism, Bryan School of Business, University of North Carolina at Greensboro, Greensboro, North Carolina, United States of America

* Machiel.bouwmans@hu.nl

## Abstract

### Background

Digital transformation (DT) involves integrating digital technologies into organizations to improve productivity, efficiency, and quality. Investing in the workforce's skillsets is essential for successful DT. However, it remains unclear which skillsets are essential.

### Objectives

This study aims to identify and define the essential skillsets needed for exploiting the full potential of DT, and to consolidate the identified skills into a comprehensive framework of DT skills.

### Method

A systematic literature review was conducted using the PRISMA approach for selecting studies. This led to the selection of 36 articles that were examined using thematic analysis for identifying and consolidating skills into a framework.

### Results

The Digital Transformation Skills Framework (DTSF) was developed, which contains six overarching skillsets and 44 underlying skills. The framework covers key skillsets in the areas of digital work, entrepreneurship, evidence-based work, collaboration, communication, and adaptation.

### Conclusion and discussion

The DTSF offers a comprehensive understanding of essential skills for today's evolving organizations, addressing a critical gap in existing literature. It is valuable for organizations and HR professionals, serving as a foundation for re- and upskilling initiatives. Ongoing

**Data Availability Statement:** All relevant data are within the manuscript and its Supporting information files.

**Funding:** This research was conducted with internal funding provided by the HU Utrecht University of Applied Sciences. The funders had no role in study design, data collection and analysis, decision to publish, or preparation of the manuscript.

**Competing interests:** The authors have declared that no competing interests exist.

research should expand the framework to include domain-specific DT skills and emerging digital technologies.

## Introduction

The rapid advancements of digital technology, including automation, artificial intelligence (AI), big data, cloud computing, robotics, and internet of things (IoT), have profoundly impacted organizations [1–3]; for instance, by enabling organizations to process, archive, and access information at an unprecedented scale, facilitating real-time data analysis, and improving productivity, efficiency, and quality [2]. These advancements drive digital transformation (DT), which comprises the process of improving organizations "by triggering significant changes to its properties through combinations of information, computing, communication, and connectivity technologies" [4, p. 121]. Organizations across sectors undergo DT as they adopt new digital technologies to redefine value propositions for stakeholders as well as optimize or develop digital strategies, structures, processes, and operations [1, 5]. This makes DT a central aspect of Industry 4.0, given that Industry 4.0 entails high digitalization and use of information technologies by organizations to adapt to rapidly changing environments and to gain competitive advantage [6, 7].

Successful DT of organizations relies heavily upon its workforce. Trenerry et al. [3] identified several people-related factors that determine the success of DT, such as employees' skillsets, perceptions, and attitudes towards technological change, team adaptability and resilience, and organizational culture. Particularly, employees' skillsets are often recognized as an important prerequisite to successful DT [8], as advancements of digital technology are shifting the skills needed in the workplace. Employees not only need digital skills, which are the abilities needed to perform and complete job tasks within digitalized work environments [3], but also additional, non-digital, skills to thrive in the context of DT [9]. Skills gaps emerge and grow, as employees increasingly do not have these essential skills required to perform their jobs in rapidly changing work environments. Notably, in 2016 the World Economic Forum [6] predicted that 35% of employees' skills would be disrupted in the upcoming five years due to digital technology advancements, and that share has risen to 44% in 2023.

Subsequently, the World Economic Forum [6] predicts that 60% of employees will require re- and upskilling training activities in the next five years, but only half have access to adequate training opportunities. There is, therefore, a great responsibility on organizations to prevent skill obsolescence and to invest in re- and upskilling activities for their workforce. To successfully do this, it is crucial to have a comprehensive understanding of essential skills for DT [2].

However, there is no consensus on which specific skillsets are needed in the context of DT. Even though there are many scientific articles and models discussing the skills needed for DT, existing frameworks do not cover all the essential skills. This is largely because extant research and frameworks have specifically focused on digital skills [e.g., 10, 11] and, as a result, overlook other crucial skills necessary for exploiting the full potential of DT [9, 12].

Therefore, we attempt to fill this gap in the current literature by making the following three contributions in this research. First, we seek to discern the skills necessary for both the digital technology component and the transformational component of DT, both of which are fundamental in the context of DT [5]. Second, recognizing that DT transcends specific sectors or professions, we pursuit to pinpoint skills that hold relevance across a wide spectrum of professions spanning various sectors. Third, we aim to consolidate the identified skills into a

comprehensive framework of DT skills, accompanied by skill definitions to mitigate conceptual ambiguity.

We pursue these contributions by conducting a systematic literature review, through which the following research questions will be answered:

1. *Which workforce skills are essential for digital transformation*?

2. *How can these essential skills be synthesized into a digital transformation skills framework*?

## Digital transformation skills

Industry 4.0 is characterized by the adoption of different interdependent digital technologies across diverse sectors, to enhance processes, decision-making, and services, changing the ways organizations operate and interact with customers, and fundamentally changing the nature of work [13]. The speed and scale of ongoing advancements of digital technologies lead to a digital disruption regarding work, because jobs or tasks are being displaced by digital innovations, and at the same time new jobs and tasks arise due to the emergence of new digital technologies [3, 14]. This makes skill disruption inevitable. For example, as digital technologies replace employees in performing certain tasks, the skills needed to perform those tasks become obsolete, and new digital technologies will create new tasks for which new skills are needed [5]. This disruption is not unexpected as industrial revolutions have historically displaced jobs and industries, necessitating new and often more complex skillsets [14].

The transformative shift of Industry 4.0 is generally viewed as a DT, which, in turn, is regarded as a socio-technical process of exploiting the potentials of digital technologies for strategic organizational purposes, often as a response to (possible) marginalization or displacement of an organization due to the digital advancements of other organizations in its sector [15]. Compared to digitization (converting from analogue to digital processes), and digitalization (using the potentials of digital technologies for mainly operational purposes), DT involves development towards direct integration of digital technologies in digital business strategies [15].

For digital strategies to be successfully implemented, organizations must undergo significant changes in structures and processes. It is often argued that an agile approach [16], or adoption of a malleable organizational design [17], is needed to respond to and leverage emerging digital technologies, as these increase the likelihood of maintaining competitiveness, exploiting new opportunities, and adeptly navigating unpredictable situations [17]. This need for agile approaches and malleable organizational designs underscores how DT differs from other organizational change processes which are often episodic and infrequent, whereas DT is ongoing, evolving, and cumulative [17]. Gong and Ribiere [1] therefore describe DT as a transformative shift, driven by innovative use of digital technology and strategic resource optimization, aiming to radically enhance and redefine value propositions for stakeholders within organizations, or even business networks, industries, or society.

Employees are regarded as one of the strategic pillars for DT [9], as their ability to adapt to DT is a critical determinant of its success [1]. When some tasks become obsolete and other tasks emerge, it is crucial that employees adopt essential skillsets for DT [18]. This will be challenging for employees for several reasons. First, digital technologies are advancing at a rapid and exponential rate, whereas employees can currently only adapt at a much slower rate, and this growing skills gap will overpower some employees if they are not acutely aware of their potential skills obsolescence [12]. Second, employees in DT contexts not only need job-specific and ever-changing technological skills, but also relevant soft skills such as collaboration skills

[12], problem solving skills, and project management skills [2]. This is affirmed by the World Economic Forum [6] which, alongside technological skills, identifies soft skills related to collaboration, communication, entrepreneurship, and self-efficacy as increasingly vital. Foerster-Paster and Golowko [19] and Ivaldi et al. [2] even argue that soft skills are more valuable for employees than digital technological skills, as they contribute to one's adaptability and flexibility, which are essential to adaptive learning in the dynamic context of DT.

Organizations bear responsibility for preparing employees for DT, by developing upskilling activities through which employees can learn new skills needed to perform new tasks, and reskilling activities through which employees learn new skills needed to perform new jobs [2, 20]. Responsibility for re- and upskilling for DT typically lies with human resource management (HRM) [7, 20–28]. However, although there is consensus that expectations towards employees are changing and that organizations need to implement re- and upskilling activities, there is hardly agreement on which specific characteristics of employees are missing specific to DT [29]. In other words, there is no consensus on which specific skills are increasingly important. Therefore, identifying these essential skills is key for re- and upskilling activities to be successful [2].

Despite the increasing number of scientific articles on the subject, there is no unequivocal answer as to which DT skills are essential. The primary reason for this is the conceptual ambiguity surrounding essential skills in increasingly digitalized workplaces. This ambiguity is the result of ongoing advancements in digital technology, which require concepts of digital technology-related skills to rapidly evolve as well [30]. Consequently, various and often overlapping skills frameworks, such as digital literacy, information literacy, and digital competence, have emerged to describe essential digital technology-related skills. Apart from this conceptual ambiguity, most of these existing skills frameworks do not fully capture the complex and multifaceted nature of DT. For instance, many skills frameworks focus exclusively on technological or digital skills, failing to explain the essential soft skills necessary to exploit the full potential of DT [9]. As DT is a fundamental organizational change process [1], a comprehensive framework of essential skills for DT should go beyond digital technology-oriented skills and include soft, transformation-oriented skills as well.

Therefore, we consider essential DT skills to be a combination of digital technology-oriented skills (hard skills) that enable the optimal use of new digital technologies in work, and transformation-oriented skills (soft skills) that enable adaptability and flexibility in the wake of changing work conditions. This is the starting point for developing a new comprehensive framework of essential skills for DT.

## Materials and methods

We conducted a systematic literature review (SLR) to identify and synthesize essential DT skills in a transparent and reliable way. SLRs are used to advance knowledge based on prior existing academic work. As with all scientific research methods, it is important that SLRs are executed in a valid, reliable, and repeatable manner [31]. Therefore, in this paper we used the PRISMA 2020 approach [32], which includes the process of identifying, screening, and including academic papers. Moreover, we used the PRISMA 27-item checklist for reporting the SLR results in a transparent and complete manner, along with information regarding the advanced Boolean search action, eligibility criteria for including academic papers, quality assessment, and the process of achieving inter-rater agreement. The process of this SLR is visualized in the PRISMA 2020 flow diagram (Fig 1) and the PRISMA Checklist adopted for this study is included as S1 Table.

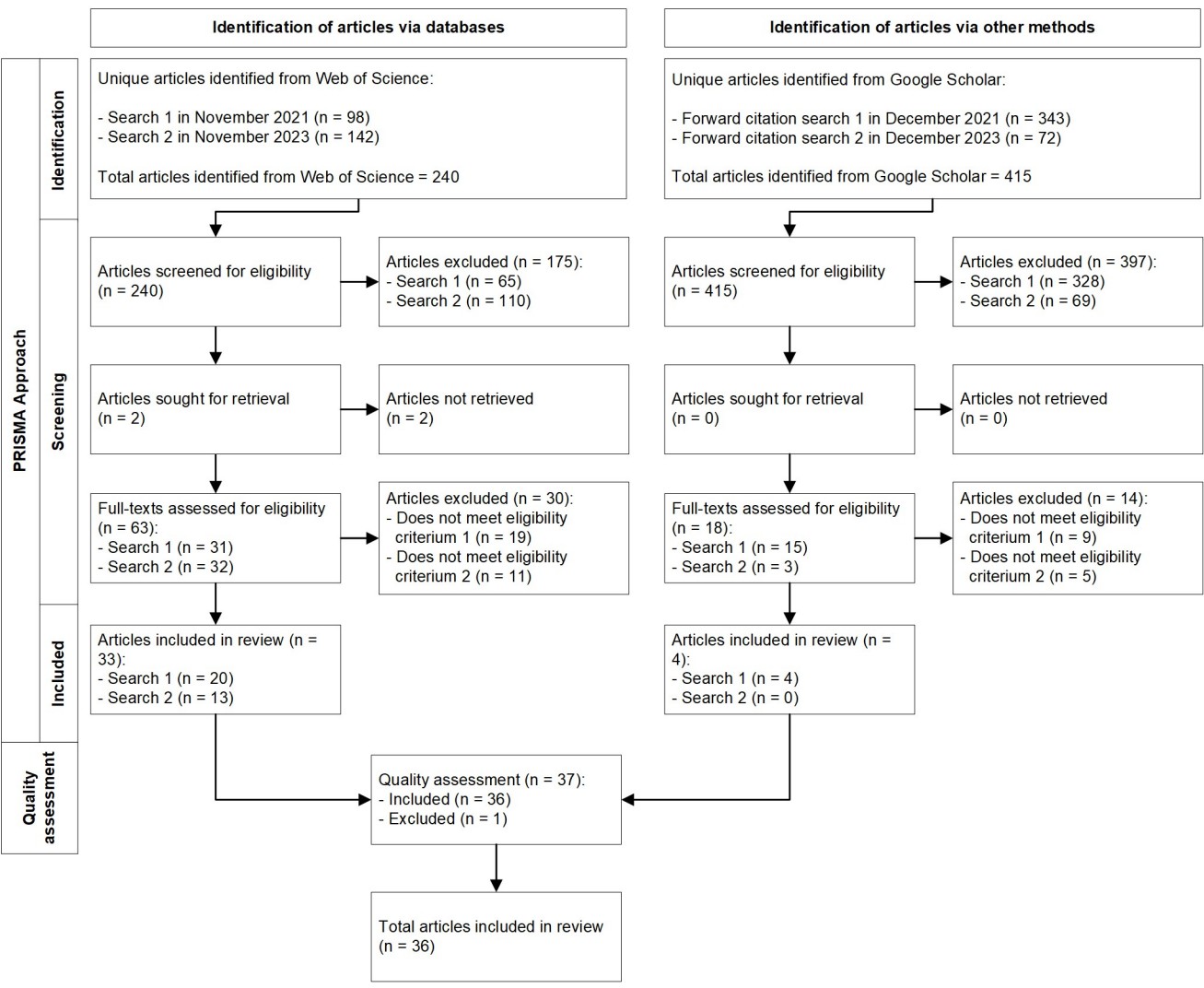

**Fig 1. PRISMA flow chart for the identification of included studies.**

## Search strategy and selection process

Our search strategy consisted of two steps: an advanced search action in Web of Science, followed by a forward citation search in Google Scholar for the selection of articles found in Web of Science. Searches were conducted in November–December 2021 and in November-December 2023.

**Step one: Selection of articles via web of science.** For the search action in Web of Science, we combined multiple search terms into one Boolean search operator. Fig 2 presents the search terms by abstract and keywords (see S1 Fig for the detailed Boolean search operator). This resulted in the identification of 240 articles. After identification, we screened the articles in two stages. In stage one, we screened the titles and abstracts of all 240 articles for the following eligibility criteria:

- Includes conceptualizations, definitions and/or measurements of DT skills or a related term.

| Abstract of the article contains: | | | | Keywords contain: |
|---|---|---|---|---|
| ('Digital transformation skills' OR 'Digital transformation competences') OR (('Digital skills' OR 'Digital competences') AND 'Transformation') OR (('Skills' OR 'Competences') AND ('Transformation' OR 'Digitalization')) OR ('21st century digital skills' OR '21st century digital competences') OR (('21st century skills' OR '21st century competences') AND ('Digital transformation' OR 'Digitalization')) | AND | ('Framework' OR 'Model' OR 'Review' OR 'Measurement' OR 'Instrument') | AND | ('Competences' OR 'Skills') |

**Fig 2. Boolean search action.**

- Includes DT skills or related terms that are not too domain-specific for one profession or sector.

- Published in a peer-reviewed journal.

- Written in English.

Three co-authors independently conducted this round of screening and discussed differences in judgment to reach absolute interrater agreement. After the stage one screening, we excluded 175 articles based on the eligibility criteria, and two articles could not be retrieved. The main reasons for exclusion were (a) the articles had no specific focus on skills for DT, (b) the articles contained only domain-specific skills for DT (e.g., for healthcare or educational professionals only), or (c) the articles focused on skills for organizations instead of employees. We followed the same process for stage two, this time reviewing the full texts of the remaining 63 articles. Reasons for exclusion of articles in this stage are included in Fig 1. This procedure resulted in the final inclusion of 33 articles from the Web of Science database.

**Step two: Selection of articles via Google Scholar.** To be included in the forward citation search in Google Scholar, the 33 selected papers from Step One needed to:

- Include rich definitions of skills or rich descriptions of indicators of skills.

- Have a minimum average of two citations per year. We deemed this minimum acceptable given that the majority of articles on this topic are recent (published since 2020) and thus have not had time to collect more citations.

Furthermore, if papers had the same authors, papers were only included in the forward citation search if they contained different frameworks or models in multiple articles.

Based on these criteria, nine articles from Step One were selected for inclusion in the forward citation search. In total these nine articles were forward-cited by 415 papers, which were screened and assessed in the same two stages as described in Step One. In stage one, one co-author screened the titles and abstracts of all 415 articles for the eligibility criteria. Since Step One had proven to produce high interrater agreement among co-authors, other co-authors were not involved in this stage for Step Two. The title-and-abstract screening resulted in the exclusion of 397 articles. The main reasons for exclusion were (a) the articles had no specific focus on skills for DT, likely due to this broader forward-citation search, (b) articles were not published in peer-reviewed journals (e.g., conference papers), or (c) articles appeared as forward-citation search result for more than one of nine articles from Step One (e.g., double hits).

In Stage Two, the full-text check on the remaining 18 articles was performed by three co-authors in the same manner as Step One. Reasons for exclusion of articles are included in Fig 1. This resulted in the inclusion of 4 articles from the Google Scholar database, for a total of 37 articles.

**Step three: Quality assessment.** In this step, we performed a methodological quality assessment of the 37 articles deemed suitable for inclusion using the JBI critical appraisal checklists for qualitative research, text and opinion papers, systematic reviews and research syntheses, and analytical cross-sectional studies [33]. One co-author conducted the assessment and then the results were discussed by all members of the research team to ensure consensus. As the critical appraisal checklists varied based on study design and methodological approach (e.g., the qualitative research checklist included 10 criteria while the cross-sectional studies checklist included 8 criteria), we converted checklist/scale scores to percentages. Quality scores ranged from 64%–100%, with an average of 94%. Using a benchmark of 80% to indicate high quality [34], only one article was excluded based on the quality assessment, resulting in a final sample of 36 articles included in our review (see S2 Table).

## Analysis and framework development

We applied template analysis to analyze the concepts used in the articles and to develop a framework of DT skills. Template analysis is a particular style of thematic analysis, in which an initial coding template is developed based on a subset of the data, which is then applied to further data and is revised and refined based on additional data, leading to a final template [35]. The concepts of the most cited article included in this SLR [9] provided our starting point for our initial coding template. We then revised and refined this initial template by comparing and adding concepts from the other included articles. To increase the reliability of our analysis, three co-authors discussed the framework and differences in judgements to reach absolute interrater agreement. This resulted in a framework containing six skillsets, of which three skillsets are further delineated into subgroups of skills, and a total of 44 different skills.

In the next step, we developed definitions for each skillset, subgroup, and skill, based on definitions, descriptions, and indicators provided in the 36 articles. Again, to increase reliability and the clarity of definitions, the three co-authors discussed the definitions to reach absolute interrater agreement.

## Results

Our analysis led to the development of the Digital Transformation Skills Framework (DTSF) (Fig 3). The results leading up to the DTSF are based on the three tables presented and discussed below, with each table serving a distinct purpose.

Table 1 presents an overview of the characteristics of the 36 included articles. Notably, it highlights the diverse terminology employed in these articles to describe essential skills for DT. Some articles primarily focused on digital technology-related hard skills, such as digital or data literacy, while others described transformation-oriented soft skills like entrepreneurial and open innovation competences. Additionally, certain articles focused on a combination of digital technology-oriented and transformation-oriented skills, employing terms like 21st century skills, transprofessional competencies, and skills pertinent to future developments such as near-future key skills, future skills, and current and foreseen skills. Despite variations in focus, all articles shared a common emphasis on essential skills for working within an increasingly digitalized work environment. Table 1 also indicates whether the articles had a broad focus on multiple professions/sectors or focused on specific professions/sectors. While the primary objective of our study was to develop a DT skills framework relevant to a wide range of

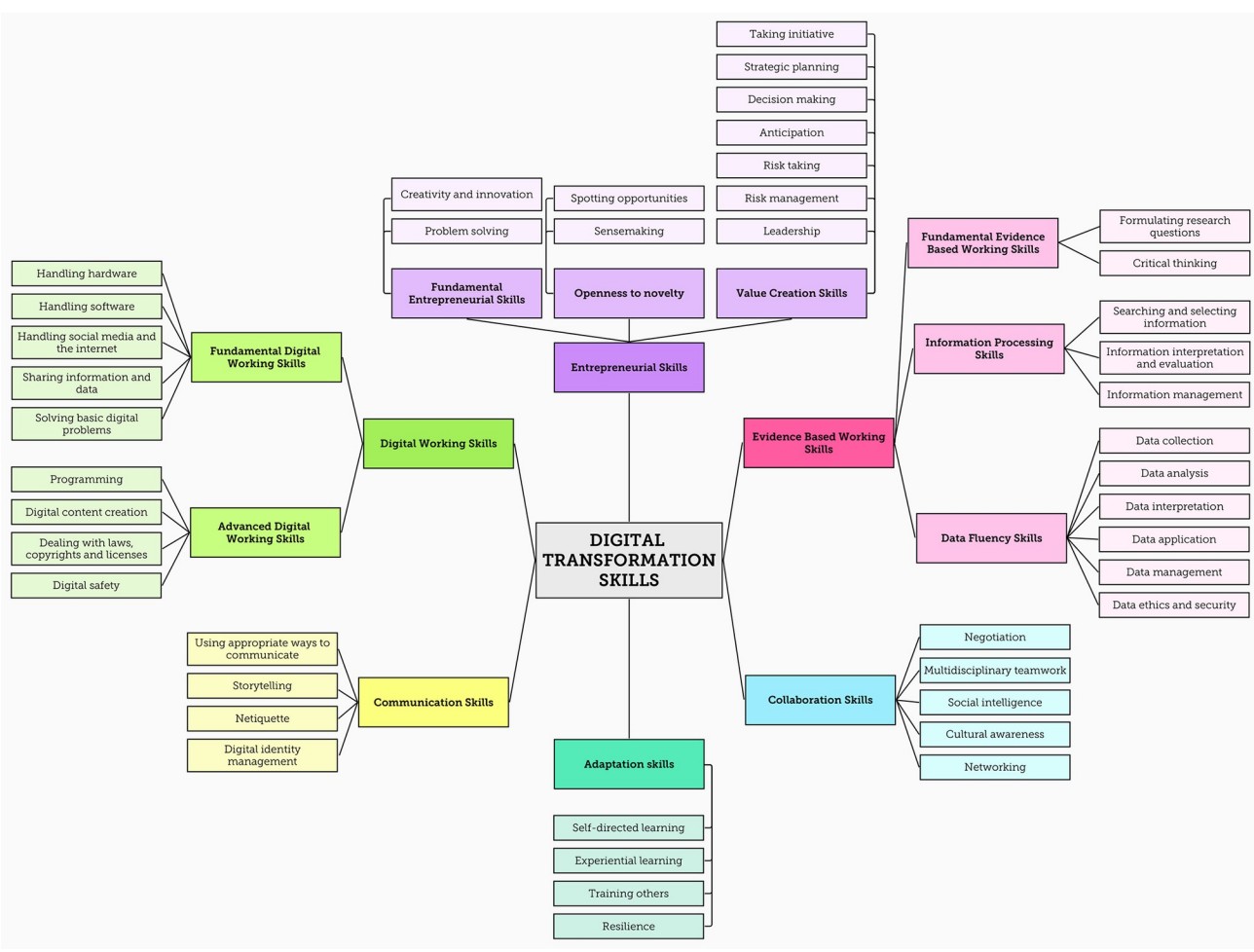

**Fig 3. The Digital Transformation Skills Framework (DTSF).**

professions, studies focusing on specific jobs or sectors were included if the skills discussed in those papers were not only domain-specific and (partially) relevant to a wider range of professions.

Table 2 presents the skillsets, subgroups, and skills that make up the DTSF. This table shows which articles, along with the terminology and concepts used in those articles, underlie the framework. In cases where different terminology was used, but similar descriptions or overlapping definitions existed, we consolidated skills and identified appropriate labels for integration into the DTSF. Table 2 also reveals instances where certain terminology used in the articles was associated with multiple skills within the DTSF. In such cases, the descriptions or definitions provided in those articles encompassed components that spanned multiple skills within the DTSF. By disaggregating these descriptions or definitions into distinct skills, our study contributes to clarifying and disentangling the conceptual ambiguity surrounding essential DT skills.

Table 3 encompasses the definitions of all skillsets, subgroups, and individual skills within the DTSF. As not all articles within the sample provided clear definitions for the skills they deemed essential, only those articles with accompanying definitions or explicit descriptions are highlighted in this table. Table 3 further aids in reducing conceptual ambiguity

**Table 1. Characteristics of included articles.**

| Article Number | Author(s)/Year | Terminology Used | Broad or Specific Focus |
|---|---|---|---|
| 1 | Akyazi et al., 2020 [36] | Near-future key skills | Specific |
| 2 | Akyazi et al., 2020 [37] | Current and foreseen skills | Specific |
| 3 | Bastidas et al., 2023 [38] | Competencies for leading digital innovation | Specific |
| 4 | Blanka et al., 2022 [39] | Employee competency | Broad |
| 5 | Busulwa et al., 2022 [40] | Digital business management competencies | Specific |
| 6 | Cetindamar Kozanoglu & Abedin, 2021 [41] | Digital literacy | Broad |
| 7 | De Souza & Debs, 2023 [42] | Workforce skills | Specific |
| 8 | Di Gregorio et al., 2019 [43] | Employability skills | Specific |
| 9 | Fan & Wang, 2022 [44] | Digital skills | Broad |
| 10 | Foerster-Pastor & Golowko, 2018 [19] | Digital and soft skills | Specific |
| 11 | García-Pérez et al., 2021 [45] | Future skills | Specific |
| 12 | Henderikx & Stoffers, 2022 [46] | Future leadership skills | Specific |
| 13 | Ivaldi et al., 2022 [2] | New competencies | Broad |
| 14 | Jardim, 2021 [47] | Entrepreneurial skills | Broad |
| 15 | Karcioğlu & Binici, 2023 [48] | Digital skills | Specific |
| 16 | Kuntadi et al., 2020 [49] | Digital skills | Broad |
| 17 | Le et al., 2020 [50] | Competency | Broad |
| 18 | Lee & Meng, 2021 [51] | Digital competencies | Specific |
| 19 | Martin et al., 2023 [52] | Skills and competencies of quality management professionals | Specific |
| 20 | Martínez-Bravo et al., 2021 [11] | Digital literacy | Broad |
| 21 | McCartney et al., 2021 [53] | HR analyst competencies | Specific |
| 22 | McPhillips & Licznerska, 2021 [54] | Open innovation competencies | Broad |
| 23 | Morris & König, 2020 [55] | Entrepreneurial competency | Broad |
| 24 | Oberländer et al., 2020 [56] | Digital competencies | Broad |
| 25 | Papamitsiou et al., 2021 [57] | Data literacy | Specific |
| 26 | Perdana et al., 2019 [58] | Digital literacy skill | Broad |
| 27 | Rayna & Striukova, 2021 [59] | 21st century skills | Broad |
| 28 | Schauffel et al., 2021 [60] | ICT self-concept | Broad |
| 29 | Schmidt et al., 2023 [61] | Big data analytics leadership competency | Specific |
| 30 | Sousa & Wilks, 2018 [62] | Critical skills | Broad |
| 31 | Trenerry et al., 2021 [3] | Skills and competencies | Broad |
| 32 | Van Laar et al., 2017 [9] | 21st century digital skills | Broad |
| 33 | Van Laar et al., 2018 [63] | 21st century digital skills | Broad |
| 34 | Vladi et al., 2022 [64] | Innovation skills | Broad |
| 35 | Wang et al., 2023 [65] | Specific leadership competencies | Specific |
| 36 | Yurinova et al., 2022 [66] | Transprofessional competencies | Specific |

surrounding essential digital transformation skills and serves as a starting point for future research on DT skills and on re- and upskilling activities.

## Digital working skills

The first skillset, digital working skills, refers to all digital technology-oriented skills needed to utilize both established and emerging digital technologies and media platforms to achieve optimal productivity and effectiveness within an increasingly digitalized work environment. The skills derived from literature are divided into two subgroups: fundamental digital working skills and advanced digital working skills.

**Table 2. Development of the digital transformation skills framework.**

| Skillsets, Subgroups, and Skills | N | Article Numbers | Terminology Used in These Articles |
|---|---|---|---|
| **Skillset 1: Digital Working Skills** | 5 | 7, 11, 12, 13, 18 | Digital competencies; Digital intelligence; Digital literacy; ICT literacy; Technical skills |
| *Subgroup 1.1: Fundamental Digital Working Skills* | 6 | 1, 2, 4, 15, 19, 31 | Basic digital skills; Basic-level digital literacy; Digital competencies; General IT skills |
| Handling hardware | 7 | 3, 5, 20, 24, 28, 32, 36 | Devices and software operations; General ICT self-concept; Handling hardware; Hardware and software operation; Technical; Understanding and using relevant equipment, hardware & devices; Working with a variety of hardware |
| Handling software | 10 | 3, 5, 8, 10, 18, 20, 24, 28, 32, 36 | Ability to utilize relevant software; Application suites; Devices and software operations; General ICT self-concept; Handling software; Hardware and software operation; Knowledge of internet & software knowledge; Technical; Understanding and using relevant equipment, hardware & devices; Working with a variety of software |
| Handling social media and the internet | 6 | 8, 10, 18, 24, 32, 36 | Handling applications; Knowledge of internet & software knowledge; Social media; Technical; Understanding of digital channels; Using the internet and social networks |
| Sharing information and data | 3 | 16, 24, 26 | Communication; Sharing data with others; Sharing information and content |
| Solving basic digital problems | 5 | 15, 16, 20, 27, 28 | Maintenance and technical troubleshooting; Problem solving; Solve problems; Solving technical problems |
| *Subgroup 1.2: Advanced Digital Working Skills* | 4 | 1, 2, 15, 31 | Advanced digital literacy; Advanced IT skills & programming; Highly specialized technical skills |
| Programming | 8 | 1, 2, 7, 10, 16, 20, 24, 27 | Advanced IT skills & programming; Programming; Programming/coding; Programming language |
| Digital content creation | 10 | 3, 5, 6, 9, 16, 20, 26, 27, 28, 36 | Content creation; Content development; Creation of digital content; Developing digital content; Digital content creation; Digital content and solutions; Developing content; General business and business function-specific DT competencies; Generate content; Integrating and re-elaborating; Production and visual design |
| Dealing with laws, copyrights and licenses | 6 | 16, 20, 24, 27, 32, 36 | Compliance with copyright; Copyright and licenses; Security and law |
| Digital safety | 14 | 1, 2, 6, 9, 13, 16, 18, 20, 24, 26, 27, 28, 31, 36 | Basic data protection knowledge; Cybersecurity; Digital safety; Digital security; Highly specialized technical skills; Protecting devices; Protecting health; Protecting health and well-being; Protecting personal data; Protecting personal data and privacy; Safe application; Safety; Security; Security and law; Using antivirus software |
| **Skillset 2: Entrepreneurial Skills** | 6 | 1, 2, 4, 13, 14, 30 | Entrepreneurship and initiative taking; Global entrepreneurial skills; Initiative and entrepreneurship; Methodological skills; Transformational or intrapreneurial competencies |
| *Subgroup 2.1: Fundamental Entrepreneurial Skills* | | | |
| Creativity and innovation | 19 | 1, 2, 4, 11, 13, 14, 16, 17, 18, 20, 22, 24, 27, 30, 31, 32, 33, 34, 36 | Applying innovation; Creativity; Creativity and innovation; Creativity skills; Curiosity and imagination; Idea generation; Innovative and adaptive thinking; Innovative capability and creativity; Innovating and creatively using technology; Innovation with digital tools; Methodological skills; Out-of-the-box thinking |
| Problem solving | 22 | 1, 2, 3, 6, 7, 8, 10, 13, 14, 16, 17, 18, 19, 20, 22, 24, 26, 30, 31, 32, 33, 35 | Analysis and problem solving; Complex problem solving; Critical analysis and judgment; Critical thinking and problem solving; Higher-order thinking and sound reasoning; Identifying needs and technological responses; Identifying problems, needs and challenges that need to be solved; Imaginative solutions; Issue & problem sensitivity; Methodological skills; Problem solving; Problem solving and critical thinking; Problem solving and decision making |
| *Subgroup 2.2: Openness to Novelty* | 1 | 14 | Skills to be focused and open to novelty |
| Spotting opportunities | 5 | 2, 4, 11, 22, 27 | Formulating strategies for taking advantage of opportunities; Identifying and defining viable market niches; Investigating opportunities; Look for opportunities; Market development; Market foresight; Opportunity assessment; Opportunity evaluation: spotting opportunities; Recognizing and taking advantage of opportunities; Spotting opportunities |
| Sensemaking | 6 | 3, 4, 11, 12, 15, 18 | Digital strategy and vision; Environmental scanning; Identifying technological trends; Pragmatic approach based on good understanding of possibilities; Sensemaking; Sensemaking skills |

*(Continued)*

**Table 2.** (Continued)

| Skillsets, Subgroups, and Skills | N | Article Numbers | Terminology Used in These Articles |
|---|---|---|---|
| *Subgroup 2.3: Value Creation Skills* | 4 | 4, 14, 20, 22 | Creation of new value; Value creation skills; Valuing ideas |
| Taking initiative | 7 | 1, 2, 4, 8, 14, 27, 30 | Entrepreneurship and initiative taking; Initiative; Initiative and entrepreneurship; Spirit of initiative; Taking initiative; Taking the initiative |
| Strategic planning | 5 | 4, 11, 14, 18, 27 | Design mentality; Personal organization; Planning and management; Planning skills; Strategic planning and evaluation; Strategic venture planning |
| Decision making | 7 | 1, 2, 7, 12, 13, 14, 18 | Critical thinking and decision making; Data-driven decision making; Decision making; Decision making abilities; Methodological skills; Problem solving and decision making |
| Anticipation | 2 | 4, 20 | Anticipation; Ethical and sustainable thinking |
| Risk taking | 1 | 20 | Risk taking |
| Risk management | 6 | 2, 4, 14, 18, 22, 27 | Coping with ambiguity, uncertainty, and risk; Coping with uncertainty, ambiguity and risk; Crisis management skills; Risk management |
| Leadership | 12 | 1, 2, 3, 4, 7, 12, 14, 17, 27, 29, 31, 35 | Inclusive style; Interpersonal sensitivity; Interpersonal skills; Leadership; Leadership and managing others; Mobilizing resources; Motivating others; Organizing and managing ability; Strategic perspective; Strategic thinking; Transformational leadership; Vision; Vision and imagination |
| **Skillset 3: Evidence-Based Working Skills** | | | |
| *Subgroup 3.1: Fundamental Evidence-Based Working Skills* | | | |
| Formulating research questions | 1 | 21 | Research and discovery |
| Critical thinking | 11 | 1, 2, 7, 8, 14, 17, 18, 30, 32, 33, 34 | Critical thinking; Critical thinking and decision making; Critical thinking and problem solving |
| *Subgroup 3.2: Information Processing Skills* | 3 | 1, 2, 6 | Complex information processing and interpretation; Information and data literacy |
| Searching and selecting information | 10 | 9, 16, 20, 24, 26, 27, 28, 32, 33, 36 | Browse, search, and filtering of data, information, and digital content; Browsing, searching, and filtering data, information, and digital content; Browsing, searching and filtering information; Information; Information management; Information skills; Process and store; Search; Selecting, evaluating and using digital content |
| Information interpretation and evaluation | 14 | 1, 2, 9, 16, 18, 20, 24, 26, 27, 28, 30, 32, 33, 36 | Ability to develop messages from digital channels; Analyze and interpret proposals and messages; Complex information processing and interpretation; Data analysis; Evaluating; Evaluating and analyzing information; Evaluating information; Evaluating data, information and digital content; Evaluation of data, information, and digital content; Fact checking skill; Information; Information management; Information skills; Process and store; Selecting, evaluating and using digital content |
| Information management | 10 | 9, 11, 16, 20, 24, 26, 27, 28, 32, 33 | Data organization; Information; Information management; Information skills; Knowledge management; Management of data, information, and digital content; Process and store; Storing and retrieving information |
| *Subgroup 3.3: Data Fluency Skills* | 9 | 3, 6, 7, 11, 13, 18, 19, 20, 21 | Analytic skills; Big data proficiency; Computational and algorithmic thinking; Computational thinking; Data fluency and data analysis; Digital literacy; Data literacy and management; Information and data literacy; Methodological skills |
| Data collection | 5 | 11, 18, 20, 21, 25 | Computational and algorithmic thinking; Computational thinking; Data collection; Data fluency and data analysis; Data mining skills |
| Data management | 11 | 2, 3, 7, 11, 18, 19, 20, 21, 24, 25, 29 | Basic technical infrastructure management; Computational thinking; Data analysis and management; Data fluency and data analysis; Data management; Data management–safe storage; Data organization; Data structuring; Lifecycle assurance and quality management; Management of data, information, and digital content |
| Data analysis | 15 | 1, 2, 3, 5, 7, 8, 11, 15, 18, 19, 20, 21, 24, 25, 29 | Advanced data analysis and modelization; Analytics and intelligence; Basic statistics; Basic statistical knowledge; Computational and algorithmic thinking; Computational thinking; Data analysis; Data analyzing; Data analysis and management; Data analysis and mathematical skills; Data analysis and data visualization; Data cleaning; Data fluency and data analysis; Data modelling; Data visualization; Pattern recognition; Quantitative and statistical skills; Statistical knowledge; Understanding and using relevant data management, data analytics, and research related methods and tools |

*(Continued)*

**Table 2.** (Continued)

| Skillsets, Subgroups, and Skills | N | Article Numbers | Terminology Used in These Articles |
|---|---|---|---|
| Data interpretation | 10 | 5, 8, 11, 18, 20, 21, 24, 25, 29, 31 | Basic statistical knowledge; Computational thinking; Data analysis; Data comprehension & interpretation; Data fluency and data analysis; Interpretation skills; Profession-related use of technology; Statistical knowledge; Synthesizing and interpreting data; Understanding and using relevant data management, data analytics, and research related methods and tools |
| Data application | 9 | 3, 5, 8, 18, 19, 20, 25, 29, 31 | Ability to create content with data; Ability to synthesize information into meaningful and actionable reports; Business analysis and management; Communication; Computational and algorithmic thinking; Data application; Data reporting; KPI development skills; Monitoring; Synthesizing and interpreting data; Understanding and using relevant data management, data analytics, and research related methods and tools |
| Data ethics and security | 2 | 3, 25 | Data ethics; Safety; Security |
| **Skillset 4: Collaboration Skills** | 19 | 2, 3, 4, 6, 7, 8, 10, 11, 12, 13, 14, 16, 20, 22, 24, 26, 27, 32, 33 | Collaboration; Collaborate through digital channels; Collaborating through digital channels; Collaborating through digital technologies; Communication; Communication and collaboration; Cooperation; Interpersonal interaction; Social skills; Support teamwork; Teamwork; Teamwork skills; Virtual collaboration; Working with others |
| Negotiation | 7 | 2, 13, 19, 20, 22, 32, 33 | Collaboration; Conflict resolution; Negotiation skills; Reconciling tensions and dilemmas; Resolving conflicts and interacting with competition positively; Social skills |
| Multidisciplinary teamwork | 2 | 2, 11 | Interdisciplinary thinking and acting; Multidisciplinarity |
| Social intelligence | 1 | 11 | Social intelligence |
| Cultural awareness | 8 | 2, 3, 11, 13, 20, 24, 26, 32 | Communication; Cultural aspects; Cultural awareness; Cultural empathy; Diversity, inclusion and teamwork; Exhibiting an informed sensitivity; Intercultural competence; Social and ethical; Social skills; Value of diversity |
| Networking | 11 | 4, 13, 24, 22, 24, 26, 27, 30, 32, 33, 34 | Collaboration; Collaboration skills; Communication; Mobilizing others; Networking; Social skills |
| **Skillset 5: Adaptation Skills** | 11 | 1, 2, 3, 12, 17, 18, 29, 30, 31, 32, 36 | Adaptive skills; Adaptability; Adaptability and adapt to change; Adaptability and continuous learning; Adaptability to new conditions; Agility and adaptability; Continuous learning; Learning ability; Lifelong learning; |
| Self-directed learning | 9 | 3, 16, 20, 23, 24, 27, 31, 32, 36 | Identification of digital competence gap; Identifying digital competence gaps; Personal and professional empowerment; Recognizing one's own knowledge gaps; Reflection; Self-direction; Self-directed learning; Self-reflection |
| Experiential learning | 4 | 4, 23, 27, 31 | Experiential learning; Informal learning; Learning through experience |
| Training others | 3 | 1, 2, 24 | Teaching and training others; Train others |
| Resilience | 5 | 4, 7, 14, 22, 31 | Motivation and perseverance; Resilience; Self-regulation and stress management; Stress resilience |
| **Skillset 6: Communication Skills** | 17 | 2, 6, 7, 9, 10, 12, 13, 14, 16, 20, 24, 26, 27, 30, 31, 32, 34 | Advanced communication skills; Communication and collaboration; Communication; Communication skills; Communication in technology-rich environments; Communication of digital content; Digital communication; Effective oral and written communication; Interacting through technologies; Social skills; Interacting through digital technologies |
| Using appropriate ways to communicate | 6 | 1, 2, 20, 24, 28, 29 | Analysis of media impact; Communicate; Communication; Communication skills; Social perceptiveness; Use of digital communication tools |
| Storytelling | 6 | 8, 14, 19, 20, 21, 29 | Clear and visual communication; Conceptualization of complex ideas in a clear and effective way; Convincing/selling ideas; Oral communication and presentation skills; Storytelling; Storytelling and communication |
| Netiquette | 7 | 3, 9, 16, 20, 24, 27, 32 | Digital empathy; Diversity, inclusion and teamworking; Ethical awareness; Ethics, fairness, and responsibility; Ethics and morals; Netiquette; Social and ethical |
| Digital identity management | 4 | 16, 20, 26, 27 | Digital identity management; Managing digital identity; Safety |

**Note:** When identical terminology appears across multiple skillsets, subgroups, or individual skills, it signifies that the descriptions or definitions associated with this terminology encompass more than one skillset, subgroup, or skill.

Table 3. Digital transformation skills framework with definitions.

**Skillset 1: Digital Working Skills (18)**
Digital working skills encompass the essential skills necessary to proficiently utilize both established and emerging digital technologies and media platforms to achieve optimal productivity and effectiveness within an increasingly digitalized work environment.

*Subgroup 1.1: Fundamental Digital Working Skills (19)*
Fundamental digital working skills encompass the generic essential skills required to proficiently operate and navigate hardware, software, the cloud, internet, and social media applications, enabling individuals to effectively carry out everyday tasks across a wide range of professional domains.

**Handling hardware (3, 6, 20, 24, 32)**
Handling hardware encompasses the proficiency to effectively manage and utilize various hardware components within the workplace. This skill involves adeptly identifying and using the functions and features of (mobile) devices to accomplish everyday tasks.

**Handling software (3, 5, 6, 20, 24)**
Handling software encompasses the proficiency to effectively manage and utilize essential software applications within the workplace. This skill involves adeptly navigating and using basic software tools; for example, for text editing, making spreadsheets, and sending emails. It includes the skill to identify and utilize the appropriate functions and features of these software applications to enhance efficiency and productivity.

**Handling social media and the internet (24, 32)**
Handling social media and the internet encompasses the proficiency to effectively utilize social media applications and leverage the internet for everyday tasks. This skill involves adeptly navigating and leveraging various social media platforms and online resources, and identifying suitable social media platforms based on their features, target audience, and functionality, enabling individuals to effectively communicate, collaborate, and engage with relevant communities and networks.

**Sharing information and data (24, 26)**
Sharing information and data involves the proficiency to effectively distribute and exchange information and data using digital applications. This skill encompasses the ability to collaboratively work with colleagues in cloud-based platforms and shared drives for synchronous online collaboration; for example, using digital tools and platforms to securely share files, documents, and resources with others, facilitating efficient communication and collaboration in real-time.

**Solving basic digital problems (15, 20, 28)**
Solving basic digital problems encompasses the proficiency to troubleshoot on a basic level to prevent technological devices and systems from experiencing breakdowns. This skill involves a foundational level of diagnosing, resolving, and preventing common issues that may arise with digital devices and systems; for instance, by making simple repairs or adjustments to restore functionality.

*Subgroup 1.2: Advanced Digital Working Skills (15, 18, 20)*
Advanced digital working skills are of growing significance in an increasingly digitalized workplace, enabling employees to excel in their roles. These skills encompass the abilities related to more advanced digital technologies, such as the ability create and manage digital content, effectively instruct programs to automate tasks, and safeguard oneself and others from digital threats, all while adhering to pertinent laws and regulations.

**Programming (7, 20, 24)**
Programming encompasses the proficiency to effectively apply at least one programming language. This skill involves developing a coherent and comprehensible sequence of instructions to solve a specific problem or perform a particular task in, for instance, AI-assisted technologies. Proficiency in programming enables individuals to automate processes, manipulate data, and create innovative software applications.

**Digital content creation (20, 26, 28)**
Digital content creation encompasses the proficiency to generate diverse forms of digital content, including but not limited to digital documents, data, videos, images, and audio. This skill involves the ability to create original content as well as edit and enhance existing content, whether self-created or contributed by others.

**Dealing with laws, copyrights, and licenses (20, 24, 36)**
Dealing with laws, copyrights, and licenses encompasses the proficiency to navigate and adhere to the legal frameworks, copyrights, and licenses relevant to data, information, and other forms of digital content. This skill involves understanding and acting upon the legal obligations and restrictions surrounding the use, distribution, and protection of digital content; for example, compliance with applicable laws, intellectual property rights, and licensing agreements when sharing digital content.

**Digital safety (9, 20, 24, 27, 28)**
Digital safety encompasses the proficiency to effectively safeguard devices and work-related or private data from unauthorized disclosure, breaches, and intrusions. This skill involves implementing robust security measures and best practices to protect digital content and sensitive information. For instance, it includes the ability to create and manage strong, unique passwords, and to employ secure authentication methods. Moreover, it encompasses protection of one's physical and psychological well-being, for instance, through the protection of one's privacy and preventing potential dangers in a digital environment.

*(Continued)*

**Table 3.** (Continued)

---

**Skillset 2: Entrepreneurial skills (4, 14)**
Entrepreneurial skills are critical for unlocking the potential of digital transformations, driving success in creating innovative and impactful ventures, products, or services that align with the needs of organizations or target populations. These skills empower individuals to identify opportunities, take calculated risks, and effectively navigate dynamic business landscapes. By utilizing entrepreneurial skills, individuals can realize objectives, contribute to the fulfillment of organizational goals, boost strategic renewal, and as such, foster growth and adaptation in an ever-evolving digital work environment.

*Subgroup 2.1: Fundamental Entrepreneurial Skills*
Fundamental entrepreneurial skills are indispensable for employees as they serve as catalysts for idea generation and problem-solving across all stages of value creation. These skills empower individuals to think creatively and navigate challenges with agility. By mastering these fundamental skills, employees are equipped to make meaningful contributions to the creation and delivery of value in diverse contexts.

**Creativity and innovation (4, 14, 17, 30, 32)**
Creativity and innovation encompass the aptitude to apply originality in thinking, offering fresh perspectives and novel approaches to address prevailing challenges or to create new ideas and opportunities. This skill involves generating new ideas or reimagining familiar concepts to devise enhanced solutions. It necessitates generating ideas, thinking beyond conventional boundaries, challenging established mental models, defying automatic thoughts, and initiating innovative products, services, or processes.

**Problem solving (3, 14, 20, 22, 24, 32)**
Problem-solving is the proficiency to comprehend complex problem situations and to develop optimal solutions from a range of alternatives. This skill encompasses the ability to acknowledge the presence of a problem, define it meticulously, and employ creative thinking to generate imaginative solutions. It involves developing, refining, and testing prototypes in an iterative way to mitigate the risk of failure and enhance problem-solving efficacy.

*Subgroup 2.2: Openness to Novelty (14)*
Openness to novelty enables employees to interpret developments and events, uncovering hidden opportunities that elude others and envisioning possible accomplishments. These skills encompass a mindset of curiosity and receptiveness to new ideas, allowing individuals to perceive and embrace emerging trends and possibilities.

**Spotting opportunities (4, 22)**
Spotting opportunities is the proficiency to identify prospects for value creation. This skill involves analyzing the social, cultural, and market developments, discerning untapped potential, and investigating options for exploitation. By spotting opportunities, individuals can seize advantageous moments, foster innovation, and drive sustainable value generation.

**Sensemaking (3, 4, 15, 18)**
Sensemaking is the proficiency to scan the business environment for digital technology trends, discern the profound meaning and significance behind digital transformations, and extract unique insights that are vital for informed decision-making; for instance, on the level of digitalization needed for the organization. This skill involves synthesizing complex information regarding digital technology trends, connecting the dots, and unraveling the underlying patterns and implications.

*Subgroup 2.3: Value Creation Skills (4, 14)*
Value creation skills are imperative for making the most of ideas and opportunities, and for devising innovative approaches to transform these ideas and opportunities into tangible value. These skills enable individuals to translate ideas and opportunities into practical prototypes. Employees can leverage their value creation skills to effectively connect the dots between ideation and realization, thereby unlocking the potential of opportunities and transforming ideas into impactful outcomes.

**Taking initiative (4, 14)**
Taking initiative is the proficiency to embrace proactivity over reactivity and to "go for it". This skill involves the willingness to take immediate action rather than waiting for events to unfold. It entails charting a deliberate course of events by developing plans, setting objectives, and working diligently towards desired outcomes.

**Strategic planning (4, 14)**
Strategic planning is the proficiency to set clear and measurable goals for an individual or an organization, creating a comprehensive plan to ensure the attainment of those goals, evaluating the implementation of the plan and making necessary adjustments, prioritizing the activities required to achieve organizational goals, adapting the plan in response to unforeseen changes, and determining the specific individuals or teams responsible for carrying out the necessary activities to achieve the desired organizational objectives.

**Decision making (12, 14)**
Decision making is the proficiency to make well-informed choices and devise optimal strategies to address the matter at hand. This skill involves gathering relevant information, prioritizing various options, and considering potential outcomes and risks. It requires critical judgment and the ability to weigh factors such as feasibility, impact, and alignment with organizational goals.

*(Continued)*

**Table 3.** (Continued)

**Anticipation (4, 20)**

Anticipation is the proficiency to develop foresight regarding the future impact of ideas and actions. This skill entails developing an awareness of how choices made today can reverberate in the days to come. It involves projecting the potential consequences and impact of different courses of action or inaction, considering both short-term and long-term outcomes.

**Risk taking (20)**

Risk taking is the proficiency to embrace uncertainty, defending unconventional or unpopular positions, and tackling challenges. This skill involves being comfortable with making mistakes and viewing them as valuable learning opportunities. It requires the courage to challenge the status quo, explore innovative approaches, and push boundaries to achieve breakthroughs.

**Risk management (4, 14)**

Risk management is the proficiency to proactively identify, assess, and mitigate potential harmful effects and adversities while seeking timely solutions to unforeseen problems. This skill involves the ability to analyze risks, uncertainty, and ambiguity, and to make decisions to prevent or minimize their impact.

**Leadership (4, 12, 14, 29, 35)**

Leadership is the proficiency to guide teams in generating original and valuable solutions. This skill includes developing a clear vision, mission, and strategy that anticipate new or future developments, and to act upon it. It also encompasses the ability to effectively manage interpersonal relationships, foster specialized knowledge, consider others' ideas, and inspire a shared commitment among followers. Combined, this involves leveraging influence to motivate individuals to surpass self-interests and align with organizational objectives.

**Skillset 3: Evidence-Based Working Skills (11)**

Evidence-based working skills are essential for employees to extract valuable insights for their organizations. These skills enable individuals to effectively locate, assess, and interpret relevant information and data, allowing for informed decision-making. By utilizing evidence-based working skills, employees can navigate through vast amounts of information, critically evaluate its credibility and relevance, and extract meaningful insights.

*Subgroup 3.1: Fundamental Evidence-Based Working Skills*

Fundamental evidence-based working skills are crucial for extracting valuable insights from data and information, as they form the foundation for critically assessing well-defined research topics.

**Formulating research questions (21)**

Formulating research questions involves the proficiency to construct testable and relevant inquiries that contribute to uncovering underlying business challenges, drivers, and trends within the organization. This skill encompasses the ability to identify key areas of inquiry, understand the organization's goals and objectives, and formulate research questions that align with these strategic priorities.

**Critical thinking (14, 32, 34)**

Critical thinking encompasses the proficiency to think for oneself, by evaluating information and ideas critically, resisting hasty conclusions and fostering a discerning mindset. This skill involves engaging in both divergent thinking, which entails exploring multiple solutions to challenges, and convergent thinking, which involves analyzing and selecting the most suitable solution. It encompasses employing reflective reasoning to assess assumptions, biases, and logical coherence in arguments and claims.

*Subgroup 3.2: Information Processing Skills (21, 32)*

Information processing skills are essential for efficiently navigating, selecting, and synthesizing multiple streams of information, enabling individuals to make informed decisions and accomplish tasks effectively. These skills encompass the ability to define precise search terms, employ effective search strategies, and access information from diverse and relevant sources. Proficiency in information processing includes evaluating the reliability, credibility, and usefulness of retrieved information, discerning between primary and secondary sources, and critically assessing the quality of information. Additionally, it involves organizing and managing digital information to facilitate retrieval and future reference.

**Searching and selecting information (3, 16, 18, 20)**

Searching and selecting information encompasses the proficiency to conduct efficient and effective searches within the vast pool of digital information. This skill involves articulating information needs, formulating clear search queries, filtering results, and devising personalized search strategies to optimize the search process. Proficiency in searching and selecting information includes familiarity with various databases, online search engines, and specialized sources relevant to the subject matter.

**Information interpretation and evaluation (18, 20, 24, 28)**

Information interpretation and evaluation encompass the proficiency to systematically analyze and interpret digital information in a discerning and critical manner. This skill involves questioning the origins, construction, and purpose of search results and digital content, fostering a skeptical mindset that seeks to uncover biases, agendas, and potential misinformation. It includes the capacity to weigh and assess the relevance of information in relation to a given context or task, employing rigorous fact-checking and verification techniques to ensure accuracy. Additionally, it involves evaluating the quality, appropriateness, reliability, and credibility of information sources, considering factors such as expertise, objectivity, and currency.

*(Continued)*

**Table 3.** (Continued)

**Information management (20, 24, 25)**

Information management encompasses the proficiency to effectively organize, store, and retrieve digital information in a structured and meaningful manner. This skill involves employing data editing methods and utilizing data descriptions, such as metadata, to facilitate efficient storage, organization, and retrieval of information. It includes developing and implementing strategies for information categorization, classification, and tagging, enabling easy navigation and access to relevant data.

*Subgroup 3.3: Data Fluency Skills (11, 18, 19, 21)*

Data fluency skills are essential for comprehending and extracting valuable insights from large quantitative data sets, enabling individuals to derive relevant and useful information for their work. This skill set encompasses the ability to extract, clean, edit, and analyze large data sets using appropriate tools and techniques. Proficiency in data fluency includes applying statistical analysis methods to uncover patterns, trends, and correlations within the data. It involves interpreting and understanding visual representations of data through data visualization techniques, enabling effective communication and decision-making. Additionally, it entails employing data management techniques to ensure data quality, integrity, and security throughout the data lifecycle.

**Data collection (25)**

Data collection encompasses the skillful process of obtaining or accessing relevant data and evaluating its quality and limitations. This skill involves identifying the appropriate sources and methods to gather data, ensuring its relevance and applicability to the desired objectives. Proficiency in data collection also includes understanding data quality metrics, such as accuracy, completeness, and reliability.

**Data analysis (3, 15, 25, 29)**

Data analysis encompasses the skillful application of basic statistical methods for analysis and forecasting, coupled with the ability to employ suitable presentation or visualization methods. This skill involves understanding and selecting appropriate statistical techniques to analyze data, such as descriptive statistics, inferential statistics, or exploratory data analysis. Proficiency in data analysis includes following a systematic and structured approach to ensure accuracy and reliability in the analysis process.

**Data interpretation (25, 29)**

Data interpretation encompasses the proficiency to make sense of data by identifying patterns, conducting hypothesis testing to validate or reject assumptions, and understanding the underlying characteristics and limitations of the data. This skill involves the ability to discern meaningful patterns and trends within the data, recognizing relationships and dependencies that can inform decision-making. It also entails understanding the inherent properties of the data, such as measurement errors, discrepancies, or missing values, and accounting for them in the interpretation process.

**Data application (3, 8, 25, 29)**

Data application encompasses the skillful synthesis of data; for instance, for defining KPI's tied to business objectives, monitoring performance against certain metrics, identifying business needs and determining solutions, and developing meaningful and actionable reports that facilitate decision-making and the evaluation of interventions. This skill involves the ability to construct clear, concise, and coherent reports or presentations that effectively communicate the key findings, insights, and recommendations derived from the data analysis process.

**Data management (3, 20, 24, 25)**

Data management encompasses the systematic handling of quantitative data throughout its lifecycle, including organization, storage, retrieval, and maintenance. This skill involves establishing efficient and logical structures to organize and categorize data, ensuring its accessibility and usability. It also entails employing data descriptions, such as metadata, to provide contextual information and enhance the understanding and interpretation of the data.

**Data ethics and security (3, 25)**

Data ethics and security involves the conscientious consideration and responsible decision-making regarding the handling, storage, and analysis of data, with a focus on protecting personal data and privacy, ensuring ethical data handling, minimizing potential harm, and complying with requirements for data handling. It also involves asking critical questions about data security measures, including how data is protected, stored, and accessed. In addition, it entails collecting and storing data in an ethical manner, respecting the privacy and rights of data subjects, and determining appropriate data retention periods.

**Skillset 4: Collaboration Skills (3, 11, 26, 32)**

Collaboration skills are essential to effectively collaborate with others from different disciplines, backgrounds, and expertise, fostering a collaborative environment for innovation, co-creation and collective problem-solving. Proficiency in collaboration skills includes the ability to engage in (digital) collaboration, effectively communicate ideas, share knowledge, and coordinate efforts towards a common goal.

**Negotiation (32)**

Negotiation encompasses the proficiency to engage in effective and constructive communication with the aim of reaching agreements and making decisions that align with a common goal, while maintaining mutual respect for all parties involved. This skill involves navigating tensions, dilemmas, and trade-offs that may arise during the negotiation process, and finding a balance between seemingly contradictory or incompatible demands. It also entails recognizing that there can be multiple valid solutions or methods of finding a resolution, and being open to exploring creative alternatives that address the interests of all parties.

*(Continued)*

**Table 3.** (Continued)

**Multidisciplinary teamwork (11)**

Multidisciplinary teamwork involves the proficiency to collaborate with individuals from diverse disciplines to collectively contribute to the development of a common, integrated, and shared mental model. Proficiency in multidisciplinary teamwork includes actively engaging in collaborative discussions, exchanging ideas, and leveraging the diverse perspectives and expertise of team members to shape a shared understanding and approach. It also entails synthesizing and integrating knowledge from different disciplines, identifying areas of overlap, and aligning goals and strategies towards a common objective.

**Social intelligence (11)**

Social intelligence encompasses the proficiency to perceive, understand, and evaluate the emotions and dynamics within a team or working group, with the aim of adapting oneself to provide the best possible work environment for peers and fostering positive interpersonal relationships. This skill involves being attuned to the emotions and non-verbal cues of others, demonstrating empathy, and effectively interpreting and responding to social signals.

**Cultural awareness (3, 20, 24, 32)**

Cultural awareness encompasses the proficiency to demonstrate an understanding, appreciation, and respect for different cultures during collaboration. This skill involves being aware of how beliefs, values, and cultural sensitivities influence the thoughts and behaviors of individuals from diverse cultural backgrounds and being familiar with cultural norms during collaborations. It also entails being sensitive to issues of prejudice, racism, and stereotypes, and actively challenging and addressing such biases.

**Networking (4, 22, 24, 34)**

Networking encompasses the proficiency to establish and nurture (online) networks of relevant partners, both within and outside one's organization, with the goal of sharing information, services, and resources, building alliances towards a common goal, and inspiring others. This skill involves actively seeking out, connecting with, and getting on board with individuals and organizations who possess complementary expertise, knowledge, or resources that can contribute to mutual growth and success.

**Skillset 5: Adaptation Skills (12, 20, 30)**

Adaptation skills are crucial for navigating the consequences of digital transformation and for thriving in a rapidly changing workplace. These skills involve the continuous modification of one's thinking, attitudes, and behaviors to effectively cope with and adapt to current and future (digital) environments. Adaptation skills enable embracing and responding to unpredictable consequences of technology, cope with changing tasks and positions, as well as navigating disruptive changes that arise from digital advancements.

**Self-directed learning (3, 20, 23, 24, 32)**

Self-directed learning encompasses the proficiency to adapt, maintain, and develop skills relevant for changing circumstances by regulating one's own progression towards self-defined learning goals. This skill involves taking ownership of one's learning journey, identifying and acknowledging personal (digital) skill gaps, formulating clear and meaningful learning goals, planning and organizing the steps required to achieve these goals, effectively managing time and effort, and taking purposeful action towards acquiring new knowledge and skills. Furthermore, self-directed learning involves engaging in reflective practices, such as self-assessment and self-evaluation, to critically analyze the quality of learning and any outputs that result from the learning experience.

**Experiential learning (4, 23)**

Experiential learning encompasses the proficiency to learn by doing, as it entails acquiring knowledge and skills through hands-on experience in the workplace. This skill is crucial for successfully engaging in more agile project work and adapting to dynamic environments. Experiential learning involves experimenting with different approaches and methods, and engaging in self-reflection, as well as collective reflection, to extract valuable lessons and identify areas for improvement. It also entails a willingness to learn from both successes and failures, understanding what works and what does not, and incorporating these insights into future actions.

**Training others (24)**

Training others encompasses the proficiency to transfer one's knowledge and expertise to others, empowering them to enhance their digital transformation skills. This skill involves teaching new programs, and effectively sharing insights, tools, and techniques to support and strengthen individuals in their professionalization.

**Resilience (14)**

Resilience encompasses the proficiency to successfully adapt to and bounce back from disturbances and challenges that threaten one's functioning. This skill involves not only the capacity to prevent or minimize the harmful effects of adversity but also the ability to overcome obstacles and remain mentally, emotionally, and physically healthy amid adverse conditions.

**Skillset 6: Communication Skills (24, 32)**

Communication skills are crucial in today's increasingly digitalized workplace, as they enable effective transmission of information and interaction with others through diverse and innovative digital communication channels. Proficiency in communication skills involves not only the ability to convey information accurately but also to ensure that the intended meaning is effectively understood by the intended recipients.

(*Continued*)

**Table 3.** (Continued)

**Using appropriate ways to communicate (20, 24)**

Using appropriate ways to communicate is the proficiency to effectively convey messages through various digital platforms. This encompasses a comprehensive understanding of how digital communication is disseminated, presented, and organized, utilizing diverse communication formats, and tailoring communication modes and strategies to cater to specific target audiences.

**Storytelling (14, 20, 29, 30)**

Storytelling is the proficiency of crafting compelling narratives that captivate and convince audiences and provide a coherent narrative thread. This skill involves showcasing the progression of interconnected ideas through the integration of diverse digital objects, including visualizations, models, and simulations. Additionally, it encompasses the ability to present arguments in a persuasive and inspiring manner; for instance, by reducing data to a key message and leaving a lasting impact on the recipients.

**Netiquette (9, 14, 32)**

Netiquette is the proficiency to exhibit socially responsible behavior in digital environments, demonstrating respect for others and their online privacy while being mindful of the potential ramifications of one's actions. This encompasses adhering to diverse rules of digital communication, treating others with respect, employing appropriate language and vocabulary, and striving to prevent misinterpretations and misunderstandings.

**Digital identity management (20)**

Digital identity management refers to the proficiency in crafting, customizing, and effectively overseeing one or multiple digital identities while safeguarding one's online reputation. This skill encompasses the ability to curate and shape digital personas across various platforms, ensuring consistency, authenticity, and privacy.

**Note:** Definitions were based on the article numbers between brackets. If no article number is mentioned, definitions were developed by the authors.

The first subgroup contains skills needed for everyday tasks common in many professions, such as *handling hardware*, *handling software* such as mail and text software, and *handling social media channels and the internet* for a given task. This subgroup also includes *sharing information and data* with others in the cloud or shared drives for synchronous online work, and a basic level of *digital problem solving* when devices or systems do not work as intended.

The second subgroup, advanced digital working skills, includes skills that are relevant for a broad array of professions, although the level of required skill mastery may very per profession. For instance, *programming* is a very relevant skill for IT professionals but is also increasingly relevant for other employees working with big data and AI technologies in data-driven decision-making processes. Similarly, *dealing with law*, *copyrights*, *and licenses*, which requires employees not only to understand them but also to be able to act in compliance with them, becomes increasingly important for a broad array of professions, yet it belongs more to core tasks of certain professions such as digital ethics officers. The same applies to the other two skills of this subgroup: *digital content creation* such as creating or editing videos, images or audio, and *digital safety skills*, which entails adequately protecting one's devices, systems, and data from disclosure.

## Entrepreneurial skills

The second skillset, entrepreneurial skills, includes transformation-oriented skills needed to fully leverage the potential of DT. The present study identifies three interconnected subgroups of entrepreneurial skills: fundamental entrepreneurial skills, openness to novelty, and value creation skills. Most articles included in this SLR include one or more skills that are part of the larger entrepreneurial skillset, such as creativity and problem solving. However, these articles often do not explicitly link these skills to entrepreneurship and neglect to describe links between these individual skills in the context of DT.

The first subgroup, fundamental entrepreneurial skills, contains skills that are essential throughout the whole value creation process. This group consists of *creativity and innovation*, needed for generating new ideas or treating familiar ideas in new ways, and *problem solving*, which not only entails recognizing and defining problems and generation solutions, but also developing, testing, and refining prototypes that generate value.

The second subgroup, openness to novelty, contains skills needed to interpret developments and events in an organizational environment, and to identify new opportunities that arise from these developments. *Spotting opportunities* to generate value by establishing new ways of connecting and combining digital technology developments and events, and *sense-making*, or the ability to determine the deeper meaning of digital transformation and create unique insights, make up this subgroup.

The third subgroup, value creation skills, consists of seven skills needed for turning opportunities and unique insights into value: (1) *taking initiative* by immediately applying ideas until a better solution is found; (2) *strategic planning* by developing, adapting, and evaluating action plans for achieving goals; (3) informed *decision making* to determine the best strategy for problem solving and value creation; (4) *anticipation* of the short- and long-term consequences and potential impact of actions; (5) *risk taking* by making mistakes and defending unconventional or unpopular opinions to tackle problems in the value creation process; (6) *risk management* via minimizing and overcoming harmful effects and adversities while searching for solutions; and (7) the *leadership* skills to guide teams in creating original and valuable solutions.

## Evidence-based working skills

The third skillset contains evidence-based working skills that are needed in a digitalized work environment with increased information and data flows. Evidence-based working skills are a combination of digital technology-oriented and transformation-oriented skills. This skillset, which consists of three subgroups, enables employees to extract beneficial insights for their organization.

The first subgroup, fundamental evidence-based working skills, is a crucial basis for the other subgroups within this skillset. The first fundamental skill is *formulating research questions*, which is the ability to identify key areas of inquiry, understand the organization's goals and objectives, and formulate research questions that align with these strategic priorities. The other fundamental skill is *critical thinking*, which is the proficiency to evaluate information and ideas critically, resisting premature conclusions, exploring multiple solutions, and supporting claims with sufficient evidence.

The second subgroup, information processing skills, is widely emphasized in the reviewed literature. This subgroup encompasses *searching and selecting information* from vast digital sources, which involves articulating information needs, developing search strategies, and filtering results. Another skill is *interpreting and evaluating information* based on relevance and quality in a discerning and critical manner; for instance, through rigorous fact-checking and other verification techniques to ensure accuracy, and through evaluation of the quality, appropriateness, reliability, and credibility of information sources. The last skill of this subgroup is *information management*, which involves organizing, storing, and retrieving digital information in a structured and meaningful manner; for example, using data editing methods or adding metadata.

The third subgroup, data fluency skills, involves the ability to comprehend large quantitative datasets and convert them into relevant insights, such as actionable reports for decision-making. The first skill in this subgroup, *data collection*, involves identifying the appropriate

sources and methods to gather data, ensuring its relevance and applicability to the desired objectives. The second skill, *data analysis*, involves the application of basic descriptive, explorative, and inferential statistical methods, coupled with the ability to employ suitable presentation or visualization methods. The skill *data interpretation* builds upon previous skills, as it involves being able to make sense of data by identifying patterns and trends, recognizing relationships and dependencies, and also understanding the inherent properties of the data, such as measurement errors and discrepancies. The next skill, *data application*, involves the ability to construct clear, concise, and coherent reports or presentations on key findings, insights, and recommendations derived from the data. The skill of *data management* is very similar to information management and involves the systematic handling of quantitative data throughout its lifecycle, including organization, storage, retrieval, and maintenance. Lastly, *data ethics and security* involves conscientious consideration and responsible decision-making regarding the handling, storage, and analysis of data, with a focus on protecting individual privacy, ensuring ethical data collection practices, and minimizing potential harm.

## Collaboration skills

The collaboration skillset is regarded as a crucial transformation-oriented skillset in the context of DT by many authors (see Table 2). Employees must master skills that enable them to (digitally) collaborate with different types of employees in agile or cross-functional teams to create value or solve problems, or to collaborate in networks to achieve common goals regarding digital transformation. In the specific context of DT, five collaboration skills are considered important. The first is *negotiation* with the aim of reaching agreements and making decisions that align with a common goal while maintaining mutual respect for all parties involved. The second skill, *multidisciplinary teamwork*, involves the proficiency to collaborate with individuals from diverse disciplines to collectively contribute to the development of a common, integrated, and shared mental model by actively engaging in collaborative discussions, exchanging ideas, and leveraging the diverse perspectives and expertise of team members. The third and fourth skills, *social intelligence* and *cultural awareness*, both entail adapting oneself to a socially and/or culturally diverse team while considering differences in the team on emotional and cultural levels; for instance, by actively challenging and addressing issues of prejudice and stereotypes, or by effectively interpreting and responding to social signals. *Networking* is the last skill that makes up the collaboration skillset, and involves employees' ability to establish networks, form alliances, and engage relevant stakeholders inside and outside one's organization to achieve shared objectives.

## Adaptation skills

Mastering the fifth skillset, adaptation skills, is essential for employees' flexibility and agility, which is key in response to rapidly changing work conditions due to DT. Adaptation skills necessitate continuously modifying one's thinking, attitudes, and behaviors to effectively navigate current and future (digital) environments, unpredictable technology consequences, and disruptive changes. As such, it is a transformation-oriented skillset in which four distinct skills are identified. Firstly, *self-directed learning* involves taking charge of one's professional development, ands proactive re- and upskilling to meet evolving organizational and environmental demands. Self-directed learners manage their own progression towards self-defined learning goals, take appropriate actions to re- and upskill, reflect upon these actions, and as such prevent skill obsolescence. Secondly, *experiential learning* encompasses the proficiency to acquire knowledge and skills through hands-on experience in the workplace. This skill is crucial for successfully engaging in more agile project work and adapting to dynamic environments, and

involves experimenting with different approaches and methods, engaging in self-reflection as well as collective reflection, and extracting valuable lessons and identifying areas for improvement. The third skill, *training others*, involves transferring knowledge and expertise to others, empowering them to enhance their DT skills. The last skill, *resilience*, is the proficiency to successfully adapt and bounce back from disturbances and challenges that threaten an employee's ability to function. This skill involves not only the capacity to prevent or minimize the harmful effects of adversity but also the ability to overcome obstacles and remain mentally, emotionally, and physically healthy amid adverse conditions.

## Communication skills

The sixth and final skillset addresses communication skills. Communication skills are deemed important transformation-oriented skills in the context of DT, as noted by most articles included in this SLR (see Table 2). In an increasingly digitalized workplace, it is important that employees master communication skills that enable them to transmit information and interact with others via appropriate and innovative communication channels. Proficiency in communication skills involves not only the ability to convey information accurately but also to ensure that the intended meaning is effectively understood by the intended recipients. Within this skillset, four skills are distinguished. The first, *using appropriate ways to communicate*, articulates that employees can effectively convey messages through various digital platforms, and develop communication strategies and formats for specific audiences. The second skill, *storytelling*, denotes that employees are proficient in crafting compelling narratives that captivate audiences and provide a coherent narrative thread using digital tools, attractive visualizations, models, or simulations, with the goal to persuade or inspire others. The remaining two skills focus on appropriate behavior in the digital environment: *netiquette*, which emphasizes socially responsible online behavior, respecting privacy, using appropriate language, and preventing misinterpretation, and *digital identity management*, which involves effectively managing multiple digital identities and communicating in alignment with each identity.

## Discussion

Our study answers the following research questions '*Which workforce skills are essential for digital transformation*?' and '*How can these essential skills be synthesized into a digital transformation skills framework*?', by developing the Digital Transformation Skills Framework (DTSF). The DTSF offers new insights on essential DT skills for employees in a broad array of professions and organizations. This insight is crucial as active engagement of employees determines the success of DT [1], but employees' required skills are rapidly changing. The skills gap that emerges is recognized as a central hinderance in this success [8]. Although many studies stress the importance of both essential digital technology-oriented (hard) and transformation-oriented (soft) skills, contemporary skills frameworks tend to focus on the former and therefore neglect the full complexity of DT [9]. The importance of transformation-oriented skills is emphasized by the World Economic Forum [6] and multiple scientific papers [e.g., 2, 19]. However, a framework that synthesizes both types of skills specifically for the DT context was missing. The added value of the DTSF therefore lies in the inclusion of both digital technology-oriented and transformation-oriented skills.

The DTSF represents the multifaceted and dynamic nature of DT through six interconnected skillsets: digital working skills, entrepreneurial skills, evidence-based working skills, collaboration skills, adaptation skills, and communication skills. The current study contributes to reducing conceptual ambiguity in contemporary literature through careful examination of terminology, descriptions and definitions used in the included articles, followed by synthesis of

this information into skills, subgroups, and skillsets. For instance, the skill *data analysis*, was identified in 15 papers that all used different terminology, such as 'data analysis and mathematical skills', 'statistical knowledge', and 'quantitative and statistical skills', and the skill *creativity and innovation* was identified in 19 articles that used 12 different terms, such as 'curiosity and imagination', 'innovative and adaptive thinking', and 'out-of-the-box thinking' (see Table 2 for a complete overview).

Moreover, our study also contributes to reducing conceptual ambiguity on essential DT skills by formulating definitions for all skills, subgroups, and skillsets in the DTSF, based on the articles included in our sample (see Table 3). It was notable that not all included articles contained complete sets of definitions for the skills included in those articles. These definitions can serve a foundation to build upon in future studies on essential DT skills.

## Practical relevance and implications

The DTSF serves as a solid basis for raising awareness on skillsets that are vital across various professions and sectors and provides a starting point for sector-wide re- and upskilling initiatives. The present study therefore calls upon organizations, particularly HRM professionals, to adapt their strategic talent management practices to the digital era and assume responsibility for re- and upskilling the workforce in essential DT skills. In the development of re- and upskilling strategies informed by the DTSF, several considerations should be taken into account.

First, prior to designing or procuring interventions such as training programs focused on specific skillsets, organizations should enrich the DTSF by providing context-specific examples of the relevance of these skills and articulating desired learning outcomes for their workforce. This will enable tailored training offerings. Second, organizations need to establish desired maturity levels for all DT skills of the DTSF. While the DTSF holds relevance for diverse professions, the study emphasizes that desired skill maturity levels may vary across specific professions or organizations. For instance, entrepreneurial skills or evidence-based working skills may not require the same level of maturity in all professions. By establishing these desired maturity levels, organizations can effectively monitor skill development and implement targeted interventions. Third, in the development of re- and upskilling strategies, organizations should consider both horizontal alignment (interconnectedness between different trainings, and connection with other interventions) and vertical alignment (alignment with the overall business strategy) to maximize potential outcomes. Lastly, organizations should periodically reassess their re- and upskilling strategies to ensure that the skills essential to their specific organizational context are given emphasis and are addressed accordingly.

## Limitations and directions for future research

The DTSF raises several follow-up research questions, partly based on the limitations of this study. The first limitation prompting future research is the potential incomplete coverage of relevant research due to our search strategy. Although we utilized Web of Science, the most widely used database of scientific publications, in combination with a forward-citation search in Google Scholar, our search methodology may have resulted in key publications being overlooked. Striving for complete coverage, future research could therefore expand the search action by including other scientific databases as well.

The second limitation prompting follow-up research is related to time sensitivity. This SLR reflects the state of knowledge up to a certain point in time, and therefore does not include skills related to very recent developments, such as generative AI. As highlighted in the Introduction, the lack of timely updated frameworks may lead to conceptual ambiguity regarding essential DT skillsets or outdated frameworks. To address this, and to provide ongoing

valuable insights for organizations, this study calls upon researchers to continually provide updates to the DTSF.

Furthermore, we have intentionally used the term 'skills' over 'competence' in this study. This decision is in line with the practical objective of our research, which is to address the re- and up-skilling challenge by pinpointing essential DT skills. We acknowledge that 'competence' is a broader concept, encompassing knowledge and attitudes as well as skills. Indeed, certain knowledge and attitudes are intrinsically linked to the essential skills outlined in the DTSF. However, our SLR indicated a degree of interchangeability between these terms, sometimes leading to conceptual ambiguity. By focusing on 'skills', we aimed to reduce potential ambiguity in this paper. Nonetheless, we recognize that re- up-skilling initiatives and programs, which aim to develop essential DT skills, inherently involve the acquisition of relevant knowledge and the shaping of appropriate attitudes. Essential DT skills are not standalone entities but are part of a larger competency framework. Consequently, future research should strive to develop this broader framework by incorporating essential DT knowledge and attitudes.

Moreover, the DTSF enables various follow-up studies that are linked to the new insights the framework provides. For instance, follow-up research on the measurement and monitoring of workforce development in DT skillsets is an important next step. The DTSF presents an opportunity to develop instruments for assessing and measuring DT skills in the workplace, and the next research step therefore involves operationalizing the DT skills outlined in this framework and creating validated and reliable measurement instruments.

Additionally, future research should focus on expanding the DTSF to address specific professions. While the strength of the current framework is that it encompasses skills applicable to a wide range of professions, it lacks domain-specific DT skills. Given the complexity of DT, the importance of employees with π-shaped skillsets is becoming increasingly apparent [67], as these skills contribute to better collaboration between people with different expertise and to innovative output [68]. The term 'π-shaped skillsets' is a metaphor to describe employees who are generalists and specialists at the same time. The two vertical bars of the π symbol reflect deep expertise, such as domain-specific skills and the evidence-based working skills of the DTSF, and the horizontal bar of the π symbol reflects the generalist skills [67, 68], such as the transformation-oriented skillsets of the DTSF. Developing expansions of the DTSF for specific professions, such as by adding a digital pedagogical skillset for teachers or a digital health skillset for healthcare professionals, therefore further enhances the understanding of profession- or sector-specific essential DT skills.

Lastly, future research should examine the development of effective re- and upskilling strategies, based on the DTSF. This research should also address strategies that consider employees' digital mindset and self-efficacy in re- and upskilling, as these factors impact their engagement and willingness to participate in such activities [69].

## Conclusion

In conclusion, the Digital Transformation Skills Framework (DTSF) offers a valuable and comprehensive insight into the essential DT skills required by employees in today's rapidly changing organizations. This framework addresses a crucial gap in existing literature by synthesizing both digital technology-oriented and transformation-oriented skills, providing a holistic understanding of essential skills related to the multifaceted nature of DT. The DTSF has practical relevance for organizations and HR professionals, serving as a foundation for re- and upskilling initiatives. Ongoing research is needed to continually update and expand the

DTSF, by addressing domain-specific DT skills and including skills related to emerging digital technologies.

## Supporting information

**S1 Table. PRISMA checklist.**
(DOC)

**S2 Table. JBI quality checklist.**
(XLSX)

**S1 Fig. Detailed Boolean search action.**
(DOCX)

## Author Contributions

**Conceptualization:** Machiel Bouwmans, Xander Lub.

**Formal analysis:** Machiel Bouwmans, Xander Lub, Marissa Orlowski, Thuy-Vy Nguyen.

**Funding acquisition:** Xander Lub.

**Methodology:** Machiel Bouwmans, Xander Lub, Marissa Orlowski, Thuy-Vy Nguyen.

**Visualization:** Machiel Bouwmans.

**Writing – original draft:** Machiel Bouwmans, Marissa Orlowski.

**Writing – review & editing:** Machiel Bouwmans, Xander Lub, Marissa Orlowski.

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
