## [Decision Letter · Decision Letter 0]

8 Apr 2024

PONE-D-24-03076Developing the Digital Transformation Skills Framework: A systematic literature review approachPLOS ONE

Dear Dr. Bouwmans,

Thank you for submitting your manuscript to PLOS ONE. After careful consideration, we feel that it has merit but does not fully meet PLOS ONE’s publication criteria as it currently stands. Therefore, we invite you to submit a revised version of the manuscript that addresses the points raised during the review process.

We look forward to receiving your revised manuscript.

Kind regards,

Tachia Chin

Academic Editor

PLOS ONE

Journal Requirements:

This research was conducted with internal funding provided by the University of Applied Sciences Utrecht.

Additional Editor Comments:

As a tip, please make a list of the criteria used to search the target paper in part of "Search strategy and selection process", such as which keywords are used?

Reviewers' comments:

Reviewer's Responses to Questions

**Comments to the Author**

1. Is the manuscript technically sound, and do the data support the conclusions?

Reviewer #1: Yes

Reviewer #2: Yes

2. Has the statistical analysis been performed appropriately and rigorously? 

Reviewer #1: N/A

Reviewer #2: N/A

3. Have the authors made all data underlying the findings in their manuscript fully available?

Reviewer #1: Yes

Reviewer #2: Yes

4. Is the manuscript presented in an intelligible fashion and written in standard English?

Reviewer #1: Yes

Reviewer #2: Yes

5. Review Comments to the Author

**Reviewer #1:** Thank you for the opportunity to review. The manuscript is well written, and the systematic literature search was conducted in line with the recommended guidelines. The methods section is written clearly, and the results are described in great detail. My only minor comments reflect the high quality of this paper.

In the introduction, it might work better if the authors define Industry 4.0 and digital transformation skills prior to introducing the research questions. Also, it would be beneficial to highlight the gaps in the current literature regarding digital transformation in order to justify the current review.

Under 'Analysis and framework development', it is unclear what the authors meant by 'interrater agreement'. Was it an absolute agreement among raters, and how many raters were involved?

**Reviewer #2:** This is a timely paper, it is well researched analysed and presented. It comes at a time when the world, both academic and industry is struggling to make sense of the rapid transformations of work as a result of digitalisation. The framework will therefore make great contribution towards more accurate understanding of industry workforce digital skills needs in order to design effective ways of developing them. The research process and associated arguments are clearly presented in an overall highly accessible writing style.

6. PLOS authors have the option to publish the peer review history of their article (what does this mean?). If published, this will include your full peer review and any attached files.

Reviewer #1: No

Reviewer #2: **Yes: **Professor Victor Gekara

---

## [Author Response · Author response to Decision Letter 0]

17 Apr 2024

We have made a diligent effort to address all received requests and comments from the editor and the reviewers. Our detailed responses to all requests and comments are included in our Response to Reviewers document, which is included with this revision.

---

## [Editor Report · Decision Letter 1]

7 May 2024

Developing the Digital Transformation Skills Framework: A systematic literature review approach

PONE-D-24-03076R1

Dear Dr. Bouwmansc,

We’re pleased to inform you that your manuscript has been judged scientifically suitable for publication and will be formally accepted for publication once it meets all outstanding technical requirements.

Additional Editor Comments:

Additional Editor Comments:

Please add references of these two papers:

10.1080/13602381.2023.2290263

10.1057/s41599-024-03003-7

Kind regards,

Tachia Chin

Academic Editor

PLOS ONE

---

## [Editor Report · Acceptance letter]

3 Jun 2024

PONE-D-24-03076R1 

PLOS ONE

Dear Dr. Bouwmans, 

I'm pleased to inform you that your manuscript has been deemed suitable for publication in PLOS ONE. Congratulations! Your manuscript is now being handed over to our production team.

Kind regards, 

on behalf of

Dr. Tachia Chin 

Academic Editor

PLOS ONE